# Scanning Electron Microscopic Analysis of Antennal Sensilla and Tissue-Expression Profiles of Chemosensory Protein Genes in *Ophraella communa* (Coleoptera: Chrysomelidae)

**DOI:** 10.3390/insects13020183

**Published:** 2022-02-09

**Authors:** Chao Ma, Yang Yue, Yan Zhang, Zhen-Ya Tian, Hong-Song Chen, Jian-Ying Guo, Zhong-Shi Zhou

**Affiliations:** 1State Key Laboratory for Biology of Plant Diseases and Insect Pests, Institute of Plant Protection, Chinese Academy of Agricultural Sciences, Beijing 100193, China; machao316@126.com (C.M.); yueyang1014@163.com (Y.Y.); 17810266106@163.com (Y.Z.); tzymail@yeah.net (Z.-Y.T.); chenhongsong2061@163.com (H.-S.C.); guojianying@caas.cn (J.-Y.G.); 2School of Plant Protection, Hebei Agricultural University, Baoding 071001, China; 3Guangxi Key Laboratory for Biology of Crop Diseases and Insect Pests, Institute of Plant Protection, Guangxi Academy of Agricultural Sciences, Nanning 530007, China

**Keywords:** *Ophraella communa*, scanning electron microscopy, antennal sensilla, chemosensory protein, tissue-expression profiles

## Abstract

**Simple Summary:**

Leaf beetle *Ophraella communa* is considered an effective biocontrol agent against the common ragweed *Ambrosia artemisiifolia*. However, there are no studies that assess the antennal sensilla and chemosensory proteins expression profiles of *O. communa*. In this study, the types and morphology of sensilla on the antennae were examined by scanning electron microscopy (SEM). The results showed that there are six types of sensilla (sensilla trichodea, sensilla chaetica, sensilla basiconica, sensilla styloconica, sensilla coeloconica, and Böhm bristles) distributed on the antennae. We also found that the expression levels of five chemosensory proteins in male and female antennae were higher than those in other tissues by quantitative real-time polymerase chain reaction. Our results lay the foundation for interpreting the olfactory functions of adult *O. communa*.

**Abstract:**

*Ophraella communa* is an efficient biocontrol agent used against the invasive weed *Ambrosia artemisiifolia*. It is an herbivorous insect that feeds on specific plants; the olfactory functions of this insects plays an important role in their search for host plants. There are no reports on *O. communa* sensilla types, morphology, or chemosensory protein (CSP) genes. In this study, we observed the external structure and distribution of antennal sensilla in adult *O. communa* antennae by scanning electron microscopy; moreover, we cloned 11 CSPs (CSP1–CSP11) and elucidated their tissue-expression profiles using quantitative real-time polymerase chain reaction. Six types of sensilla were identified: sensilla trichodea (including two subtypes), sensilla chaetica, sensilla basiconica (including two subtypes), sensilla styloconica, sensilla coeloconica, and Böhm bristles. Both male and female antennae had all six types of sensilla, and no sexual dimorphism was noted in sensillar types or distribution. We also found that the expression levels of CSP2, CSP3, CSP4, CSP6, and CSP7 in male and female antennae were higher than those in other tissues, which suggests that these five CSPs may be related to olfactory function in *O. communa*. Ultimately, our results lay the foundation for interpreting the olfactory functions of adult *O. communa*.

## 1. Introduction

Olfaction plays a vital role in various insect behaviors, such as searching for host plants and mates and avoiding natural enemies [1]. Insect antennae, which are considered the main organs that perform olfactory functions in insects, have various types of sensilla distributed on them. Sensilla are specialized structures of the epidermis that can detect complex environmental stimuli (smell, taste, touch, and proprioception, as well as thermo- and hygro-reception) [2,3]. According to cuticular morphology, sensilla are subdivided into several types: sensilla chaetica (SCH), sensilla trichodea (ST), sensilla basiconica (SB), sensilla styloconica (SST), and sensilla placodea [4,5]. In addition to the antennae, sensilla are also distributed in other tissues, such as maxillary palps, labial palps, and ovipositors [6].

Throughout long evolutionary periods, insects have acquired a sensitive olfactory system, requiring a variety of proteins to cooperate. Among them, chemosensory proteins (CSPs) are extremely concentrated in sensilla lymph, serving as transporters that recognize and transfer odorants through the sensilla lymph to the odorant receptor, which is a necessary process in the olfactory system. CSPs are small and highly soluble proteins, and large number of studies have demonstrated that they can transfer various small molecules, such as odorants and pheromones to a given odorant receptor [7,8,9]. Previous studies have reported that CSPs are also expressed in non-chemosensory tissues; moreover, the functions of most CSPs are unknown. Thus, the olfactory and non-olfactory functions of CSPs need further investigation.

*Ophraella communa* LeSage (Coleoptera: Chrysomelidae), which originated in North America, is considered an effective biocontrol agent against the common ragweed *Ambrosia artemisiifolia* L. (Asteraceae) [10,11]. Both *O. communa* larvae and adults feed on the leaves of *A. artemisiifolia* and significantly suppress wild populations of *A. artemisiifolia*. *O. communa* is an herbivorous insect that feeds on specific plants and has a powerful ability to search for the host plant. Zhou et al. [12] and Ma et al. [13] have confirmed that olfaction plays a vital role in the host plant searching and mating processes of *O. communa*; in addition, CSPs of *O. communa* have been identified by transcriptome analysis [14].

However, to the best of our knowledge, there are no studies that assess the antennal sensilla or CSP expression profiles of *O. communa*. Thus, in the present study, we first used scanning electron microscopy (SEM) to observe *O. communa* antennae sensilla morphology and ultrastructure. In addition, we cloned 11 CSPs and examined their expression profiles using quantitative real-time polymerase chain reaction (RT-qPCR). Ultimately, this study aimed to improve the understanding of olfactory mechanisms in adult *O. communa* and to provide guidance for improving the future biological control of *A. artemisiifolia*.

## 2. Materials and Methods

### 2.1. Insects

Live *O. communa* beetles were obtained from the Institute of Plant Protection, China Academy of Agriculture Sciences. The beetles were reared on *A. artemisiifolia* plants in a laboratory, at a temperature of 26 ± 1 °C, a relative humidity of 75%, and a photoperiod of 14 h light and 10 h dark.

### 2.2. Scanning Electron Microscopy

The three-day mature male and female *O. communa* heads (incl. antennae) were excised as samples to be used for SEM. The heads were first cleaned using an ultrasonic cleaner for 30 s. Thereafter, the samples were dehydrated using an ascending series of ethanol concentrations (70%, 80%, 90%, and 100%) for 25 min per treatment. After air-drying for 24 h, the specimens were mounted onto SEM stubs using a double graphite adhesive tape, coated with gold in a sputter coater, and examined by SEM using a Hitachi S-3400N (Hitachi, Tokyo, Japan) at 10 kV and 3 KV. We followed the terminology provided by Zhang et al. [6] and Zhang et al. [15] to classify sensilla types. In addition, the morphological data of antennae and sensilla were collected by a SEM measurement system. For antennal morphology, we measured the antennal length and width of at least five antennae; for sensilla morphology, we measured the length and width of at least 10 sensilla for all types of sensilla. Using the pictures taken, we roughly counted the number of various sensilla and used different numbers of plus signs to represent four densities (+, slight; ++, less dense; +++, dense; ++++, extremely dense).

### 2.3. CSP Cloning and Tissue-Expression Profiles

Total RNA was extracted from 200 antennae using TRIzol Reagent (Invitrogen, Waltham, MA, USA), following the manufacturer’s protocol. The first strand of complementary DNA (cDNA) was synthesized from 1 mg of total RNA using a first-strand cDNA Synthesis Kit (Transgen Biotech, Beijing, China). The synthesized cDNA was stored at −20 °C until later use. Based on previous transcriptome data [16], 11 pairs of degenerate primers (Table 1) were designed to amplify the CSP nucleic acid sequences. Polymerase chain reaction (PCR) was performed using the following thermal program: 94 °C for 5 min, 35 cycles of 94 °C for 30 s, 48 °C for 30 s, and 72 °C for 30 s, followed by one cycle at 72 °C for 10 min. The PCR product was purified using the Monarch gel extraction kit (New England Biolabs (Beijing), Beijing, China), cloned into a Trans1-T1 clone vector (Transgen Biotech, Beijing, China) and sequenced (Sangon Biotech, Shanghai, China).

For tissue-expression profiles, 12 different tissue types (dissected from 50 individuals) were collected from virgin adults of both sexes within 3 days of adult eclosion: male antennae (M-A), female antennae (F-A), female ovaries (OV), male testis (TE), female heads (F-H), male heads (M-H), female legs (F-L), male legs (M-L), female thoraxes (F-T), male thoraxes (M-T), female wings (F-W), and male wings (M-W). Total RNA was then isolated from the 12 different tissue types, as described above. The concentration of each RNA sample was standardized to 1 µg/µL, and the cDNA was synthesized using the First-Strand cDNA Synthesis Kit for RT-qPCR (Transgen Biotech, Beijing, China), according to the manufacturer’s protocol. Ribosomal protein (RL19) was used as an internal control. RT-qPCR was performed using an ABI 7500 (Thermo Scientific, Waltham, MA, USA) with TransStar Tip Top Green qPCR Supermix (Transgen Biotech, Beijing, China). The PCR reaction conditions were 30 s at 94 °C, 40 cycles of 94 °C for 5 s, and 60 °C for 34 s. The RT-qPCR primers were designed using Primer Premier 5.0 (PREMIER Biosoft International, Corina way, Palo Alto, CA, USA), and the primer specificity and efficiency were validated prior to gene expression analysis. The RT-qPCR primer sequences are shown in Table 1. Finally, each RT-qPCR reaction was performed using three technical replicates and three biological replicates.

### 2.4. Data Analysis

RT-qPCR data were analyzed with SAS 9.0 (SAS Institute Inc., Cary, NC, USA), following the 2^−ΔΔCT^ method. Differences among treatments were evaluated by analysis of variance (ANOVA) using the Fishers Least Significant Difference (LSD) test at a significance level of *p* < 0.05. Figures were made using OriginPro 9.1 (Northampton, MA, USA). Finally, the similarity of CSP amino acid sequences were assessed using the DNAMAN software v9.0 (Lynnon Biosoft, Vaudreuil-Dorion, QC, Canada).

## 3. Results

### 3.1. Antennae Description and Sensilla Identification

*O. communa* had a filiform type of antenna, which consisted of a scape, a pedicel, and a flagellum of nine flagellomeres (Figure 1A). No difference was found in the length of segments between males and females (Table 2). All antennal segments were found to exhibit sensilla; however, the flagellum was shown to be an extremely important segment due to the numerous sensilla attached to it (Figure 1B–F).

#### 3.1.1. Sensilla Trichodea (ST)

ST were the most frequent type of sensilla and were found in all the *O. communa* antennal segments. ST had long hair-like structures, sharp tips, and bent shafts (Figure 2A). ST were further divided into two subtypes, according to their lengths, socket types, and wall surfaces.

##### Sensilla Trichodea 1 (ST1)

ST1 were long, thin, hair-like sensilla, tapering to a fine point (Figure 2A). Some cuticular pores were scattered around the ST1 sockets (Figure 2B). ST1 were situated in sockets and lay almost parallel to the cuticles. There were no pores on the ST1 surface, whereas the walls of ST1 had longitudinal grooves (Figure 2C,D). Male ST1 measured 43.45 ± 0.88 μm in length and 2.35 ± 0.09 μm in width, and female ST1 measured 42.63 ± 1.09 μm in length and 2.50 ± 0.08 μm in width (Table 3). The density of a ST1 is presented in Table 3.

##### Sensilla Trichodea 2 (ST2)

The morphology of ST2 was similar to that of ST1, despite it being shorter and thinner than ST1 (Figure 2A,F). ST2 were also situated in sockets, and ST2′s surfaces had small pores and declined at 30° angles relative to the cuticle surface (Figure 2E). Male ST2 measured 22.28 ± 0.56 μm in length and 1.88 ± 0.04 μm in width, and female ST2 measured 22.65 ± 0.48 μm in length and 2.11 ± 0.04 μm in width (Table 3). The density of ST2 is also presented in Table 3.

#### 3.1.2. Sensilla Chaetica (SCH)

SCH had a blunt tip and were slightly longer and straighter than ST1. SCH had deep curved shafts with no pores on the side wall and were set in wide sockets at 60° angles relative to the cuticle surface (Figure 2F–I). Similar to ST, pores were scattered next to the sockets of the SCH. SCH were sparsely distributed around each segment circumference, and their density increased at the distal portion of flagellomere. Male SCH measured 47.07 ± 1.48 μm in length and 2.74 ± 0.08 μm in width, and female SCH measured 48.47 ± 1.85 μm in length and 3.02 ± 0.07 μm in width (Table 3). The density of SCH is also present in Table 3.

#### 3.1.3. Sensila Basiconinca (SB)

SB were short, straight, and pointed cones. Based on morphological differences, two SB subtypes were identified:

##### Sensila Basiconinca 1 (SB1)

SB1 belonged to the longer and thinner SB subtype, resembling a standard pointed cone (Figure 3A). A few SB1 had a hole near their bottoms (Figure 3B). SB1 were set in raised circular sockets and declined at 30° angles relative to the cuticle surface (Figure 3C). Small pores were distributed on shaft side wall of SB1 (Figure 3D). Male SB1 measured 11.83 ± 0.21 μm in length and 1.45 ± 0.03 μm in width, and female SB1 measured 13.27 ± 1.40 μm in length and 1.60 ± 0.03 μm in width (Table 3). The density of SB1 is also presented in Table 3.

##### Sensila Basiconinca 2 (SB2)

SB2 were set in raised circular sockets and declined at 60° angles relative to the cuticle surface. SB2 were shorter than SB1 but thicker than SB1, and SB2 did not appear to be as sharp as SB1 (Figure 3E). Similar to SB1, small pores were distributed on the wall of SB2 (Figure 3F). Different from that of SB1, mainly distributed on the surface of antennae, the top parts of flagellomeres 4–9 displayed centrally distributed pit areas of SB2. Male SB2 measured 10.85 ± 0.28 μm in length and 1.81 ± 0.06 μm in width, and female SB2 measured 10.70 ± 0.27 μm in length and 1.80 ± 0.06 μm in width (Table 3). The density of SB2 is also presented in Table 3.

#### 3.1.4. Sensilla Coeloconica (SCO)

The appearance of the SCO can be likened to pointed nails sitting on raised sockets on the antennal surface (Figure 3G,H). They were found to be either perpendicular to or angled toward the antennae surface. SCO were highly distinctive because of the vertical stripes on their tips, which resembled a finger or unopened flower (Figure 3G). The distribution position of SCO was completely consistent with that of SB2 (Figure 3E). Male SCO measured 5.03 ± 0.32 μm in length and 1.28 ± 0.03 μm in width, and female SCO measured 4.99 ± 0.32 μm in length and 1.24 ± 0.03 μm in width (Table 3). The density of SCO is also presented in Table 3.

#### 3.1.5. Sensilla Styloconica (SST)

SST were cone-like pegs situated on raised sockets, almost perpendicular to the antennae surface. They had a multi-pores wall and there was a hole on their tips (Figure 3H,I). The distribution of SST was found to be similar to that of SCO (Figure 3E). Male SST measured 3.62 ± 0.19 μm in length and 1.13 ± 0.04 μm in width, and female SST measured 4.08 ± 0.23 μm in length and 1.12 ± 0.05 μm in width (Table 3). The density of SST is also presented in Table 3.

#### 3.1.6. Böhm Bristles (BB)

BB were only distributed on the bottom of the scapes and pedicels (Figure 4A,B). They sat in small sockets and declined by 60–90° relative to the cuticle surface. Böhm bristles (BB) were spin-like sensilla with smooth walls and sharp tips (Figure 4C,D). Male BB measured 7.16 ± 0.43 μm in length and 1.30 ± 0.05 μm in width, and female BB measured 6.46 ± 0.34 μm in length and 1.10 ± 0.09 μm in width (Table 3). The density of BB is also presented in Table 3.

### 3.2. Chemosensory Proteins (CSPs) Clone and Tissue-Expression Profiles

#### 3.2.1. CSPs Cloning and Similarity Analysis

Based on the previous *O. communa* antennal transcriptome data [16], we cloned 11 CSP genes from *O. communa* (CSP1-CSP11, GenBank number: OL456274-OL456284). The 11 CSP sequences ranged from 118 to 261 amino acids. Pairwise comparison of all CSP gene sequences showed that their similarity was between 48.18% (between CSP2 and CSP3) and 5.38% (between CSP3 and CSP11) (Table 4).

#### 3.2.2. CSP Tissue-Expression Profiles

The expression levels of 11 CSPs in M-A, F-A, OV, TE, F-H, M-H, F-L, M-L, F-T, M-T, F-W, and M-W were measured by RT-qPCR. The RT-qPCR results showed that CSP4 and CSP6 were specifically expressed in M-A and F-A, and there was no difference between expression levels in M-A and F-A. CSP1, CSP2, CSP3, CSP7, and CSP9 in M-A and F-A had relatively high expression levels but were also expressed in other tissues, such as the head, wing, and leg. In addition, expression levels of CSP1, CSP3, and CSP7 were significantly higher in M-A than in F-A. CSP5 had the highest expression level in the head, while CSP10 and CSP11 had a relatively wide range of expression patterns. The expression level of CSP8 in the OV was significantly higher than that in other tissues (Figure 5).

## 4. Discussion

Insects rely mainly on their antennae to identify various environmental stimuli, which guide them in their life activities. Sensilla distributed on the antennae play a vital role in the perception of odorants. In addition, some insect CSPs are involved in the physiological process of identifying diverse odorants. Therefore, in the present study, we assessed the morphology and distribution of six types of antennae sensilla through SEM, cloned 11 CSPs identified in the antennae transcriptome, and elucidated their tissue expression profiles.

ST were the most common and widely distributed sensilla on the antennae of *O. communa* and they appeared to be dense on the ninth flagellomere. The same distribution pattern was also found on the antennal surfaces of other Chrysomelidae insects [6]. More specifically, the same two ST subtypes found on the antennae of *O. communa* in this study were also found on *Tribolium castaneum* [17] and *Chrysolina aeruginosa* [6], much than *Phyllotreta striolata* [15], *Hippodamia variegata* [18], and *Tetropium fuscum* [19], but less than *Euplatypus parallelus* [20] and *Ips typographus* [21]. Numerous studies have reported that ST are mainly involved in the perception of sex pheromones [22,23]. In addition, Saïd et al. [23] reported that ST may also be involved in the perception of host plant odors. Furthermore, some studies demonstrated that ST can perform both mechano- and chemosensory functions based on their external and internal structures [24,25,26]. In the present study, ST1 located in a movable socket, and there were no pores on its wall, suggesting that ST1 may perform mechanical functions (Figure 2A,C). While ST2 had a multi-pores wall, this allowed them to have the olfactory function of recognizing odor molecules (Figure 2E). Furthermore, scattered pores were distributed around the ST sockets (Figure 2B). Based on previous studies, the pore may secrete enzymes, which can degrade molecules to prevent from overloading the antennal sensilla [22,26,27]. Therefore, the functions of those pores in *O. communa* remain to be elucidated.

Only one type of SCH was found on the antennae of *O. communa* in our study, which is similar to that found for *P. striolata* [15] but less similar to those found for *C. aeruginosa* [6], *H. variegate* [18], *E. parallelus* [20], *Octodonta nipae* [28], and *I. typographus* [21]. SCH are the longest sensilla on the *O. communa* antennal surface, standing straight and stretched out on each segment circumference (Figure 2F,G). In addition, no pores were found on the wall of SCH (Figure 2H,I). The description of SCH morphology and location presented here is similar to that found for other coleopteran insects. Because of their similar morphology, SCH may be recognized as ST in some insects [26]. The length and location of SCH make it easy for them to come into physical contact with the surroundings during tactile sensation, and thus, SCH are generally regarded as tactile receptors. SCH were found to be distributed particularly highly on all antennal segments, and their density increased at the distal portion of flagellomere, making it easy for them to come into contact with various substrates. 

Two types of SB were found on the antennae of *O. communa* in the present study, which is similar to the number of SB types for *O. nipae* [28] and *T. fuscum* [18] and less than the number of SB types for *C. aeruginosa* [6], *T. castaneum* [17], and *P. striolata* [15]. These two types of SB are also common in other coleopteran insects. SB are generally presumed to be olfactory receptors; for example, the aggregation pheromone of *Locusta migratoria* elicits responses, specifically from SB [29]. The types of SB found in *O. communa* was less than that in most coleopteran insects; this may arise from the fact that *O. communa* is an oligophagous insect and its SB degraded during evolutionary processes. However, both SB1 and SB2 had multi-pore walls, indicating that both SB1 and SB2 had olfactory functions (Figure 3D,F). Compared with SB1, SB2 appear to play a more prominent role in *O. communa* olfaction because there is a centrally distributed SB2 pit area (Figure 3E). Similarly, *P. striolata* have been found to have sunken regions with densely distributed SB [15].

SCO and SST have the same distribution area on the *O. communa* antennal surface. SCO have been found in *H. variegata* [18], *I. subelongatus* [30], and *C. aeruginosa* [6], and SST have been found in *E. parallelus* [20] and *C. aeruginosa* [6]. Generally, these sensilla are regarded as sensors that recognize humidity and temperature changes [31,32]; however, Altner et al. [33,34] reported that hygro- and thermo-receptive sensilla do not belong to SCO or SST in *Periplaneta americana*. This difference may be due to the long-term evolution of various insects. In the present study, no pores were found on the wall of SCO, while SST had a multi-pores surface (Figure 3G,I), indicating that SCO belonged to non-olfactory sensilla and SST had the olfactory function of recognizing odorants.

The morphology of Böhm bristles on *O. communa* appeared to be similar to that of other coleopteran insects, such as *P. striolata* [15], *C. aeruginosa* [6], and *H. variegata* [18]. In addition, Böhm bristles were found to be distributed on the articulation between the head and scape (Figure 4A) and between the scape and pedicel (Figure 4B), which is identical to findings for other insects [31]. BB had a smooth wall and no pores distributed on it (Figure 4C,D). Finally, according to previous studies, Böhm bristles are considered to be proprioceptors that perceive antennal movement and position [35,36,37].

SPs belong to an ancient and highly conserved protein family. In this study, sequences similarity analysis showed that the similarity between CSP2 and CSP3 reached 48.18%. Numerous studies have demonstrated that some CSPs have high concentrations in sensilla lymph, playing an important role in transporting odors to their receptors [38,39]; however, many CSPs are also widely expressed in other non-chemosensory tissues, such as legs and wings [40,41]. In the present study, we found that CSP2, CSP3, CSP4, CSP6, and CSP7 expression levels in M-A and F-A were higher than those in other tissues (Figure 5), indicating that these five CSPs may have an olfactory function in *O. communa*. In addition, expression levels of CSP1, CSP3, and CSP7 in M-A were significantly higher than those in F-A (Figure 5), suggesting that these three CSPs may be involved in pheromone recognition in males. We also found that CSP8 was highly expressed in the OV, which suggests a high probability of CSP8 having a role in the reproduction process. Similarly, Ma et al. [13] and Zeng et al. [42] reported a high CSP expression in ovaries, which mediated reproduction in *O. communa* and *Bemisia tabaci*, respectively. Finally, CSP1, CSP3, CSP5, CSP9, CSP10, and CSP11 had varying expression patterns, suggesting that they are involved in olfactory or non-olfactory functions, and as such, their exact functions need further investigation. What we have done in this study lays the foundation for interpreting the olfactory functions of adult *O. communa*.

## Figures and Tables

**Figure 1 insects-13-00183-f001:**
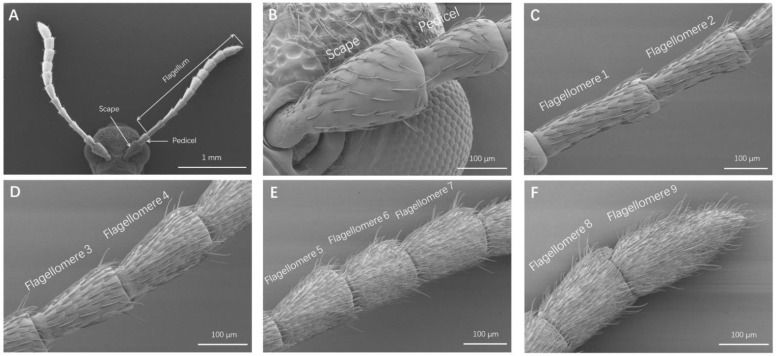
*Ophraella communa* antennae. (**A**) Overview of the female antenna showing the scape, pedicel, and flagellomeres. (**B**) Scape and pedicel. (**C**) Flagellomere 1, 2. (**D**) Flagellomere 3, 4. (**E**) Flagellomere 5–7. (**F**) Flagellomere 8, 9.

**Figure 2 insects-13-00183-f002:**
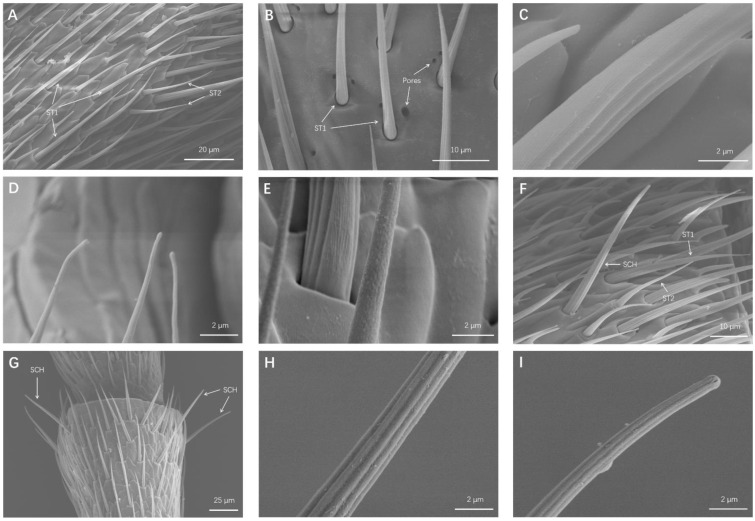
SEM micrographs of sensilla trichodea (ST) and sensilla chaetica (SCH). (**A**) ST1 and ST2 distributed on a flagellomere. (**B**) Pores were scattered around the socket of ST1. (**C**) Magnification of curved wall of ST1. (**D**) Magnification of tip of ST1. (**E**) Magnification of pores on wall of ST2. (**F**) SCH, ST1, and ST2 distributed on a flagellomere. (**G**) SCH sparsely located around the top of a flagellomere. (**H**) Magnification of side wall of SCH. (**I**) Magnification of blunt tip of SCH.

**Figure 3 insects-13-00183-f003:**
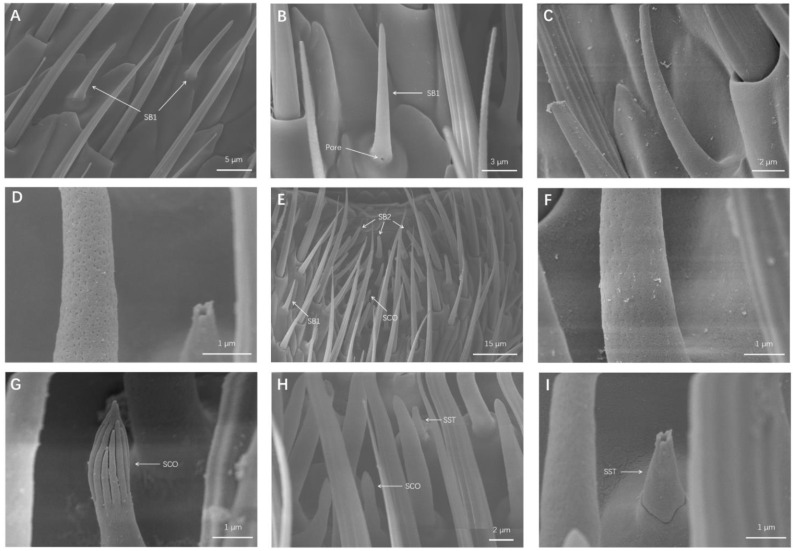
Details of sensilla basiconica (SB), sensilla coeloconica (SCO), and sensilla styloconica (SST). (**A**) SB1 distributed on a flagellomere. (**B**) A few SB1 that have a small pore on the bottom pedestal. (**C**) Magnification of SB1. (**D**) Magnification of small pores on the wall of SB1. (**E**) SB2 centrally distributed on the top section of flagellomeres 4–9, and SCO located in a SB2 densely distributed area. (**F**) Magnification of small pores on wall of SB2. (**G**) Magnification of SCO. (**H**) SST located next to SCO in a SB2 densely distributed area. (**I**) Magnification of SST.

**Figure 4 insects-13-00183-f004:**
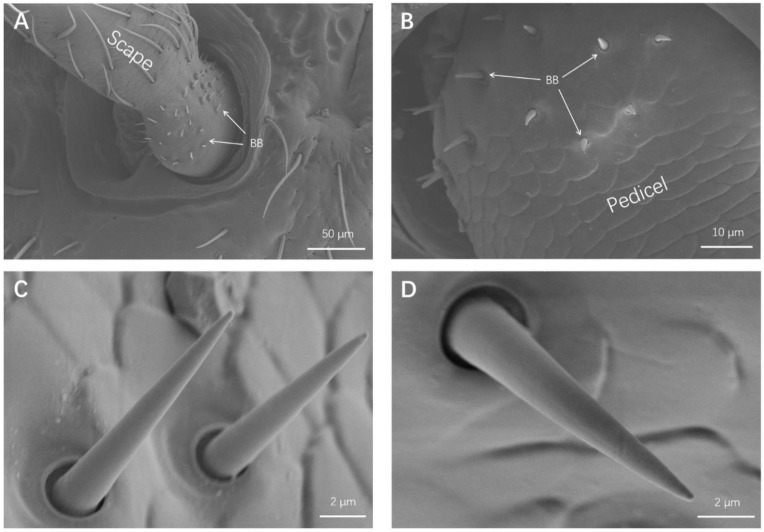
Details of sensilla Böhm bristles (BB). (**A**) BB on the joint membranes between the head and the scape. (**B**) BB on the joint membranes between the scape and the pedicel. (**C**) Magnification of BB on the scape. (**D**) Magnification of BB on the pedicel.

**Figure 5 insects-13-00183-f005:**
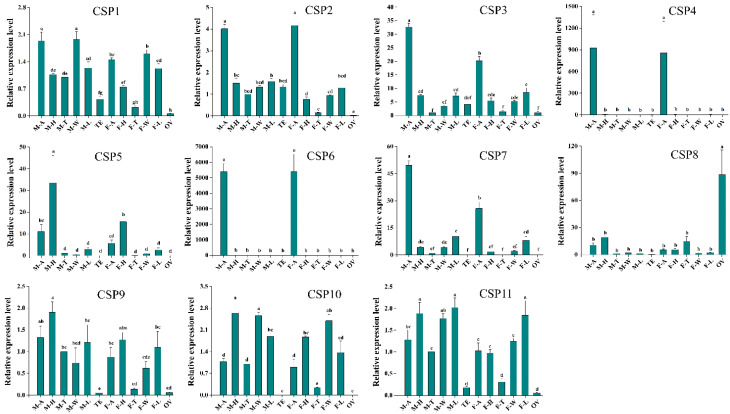
CSP expression patterns in different *O. communa* male and female tissues. All expression fold changes are related to male thoraxes. M-A, male antennae; M-H, male heads; M-T, male thoraxes; M-W, male wings; M-L, male legs; TE, male testis; F-A, female antennae; F-H, female heads; F-T, female thoraxes; F-W, female wings; F-L, female legs; OV, female ovaries. Bars with the same letters are not significantly different from each other (*p* > 0.05; LSD test). Each point represents the mean ± SE.

**Table 1 insects-13-00183-t001:** List of designed primers for polymerase chain reaction (PCR) and quantitative real-time polymerase chain reaction (RT-qPCR) of 11 *Ophraella communa* chemosensory proteins (CSPs).

Primer Name	Primer Sequence-F (5′-3′)	Primer Sequence-R (5′-3′)
**Clone primers**		
CSP1	ATGGTACCGTTAATCTGTG	CTATGTTCTGTTAAACCGGA
CSP2	ATGAATCGTTGTGGTTTGTC	TTAAATGGAGGAGTCGTTGA
CSP3	ATGAATAGTTTTTATTTATCG	TTAAATGCTCGTATCTTTAAG
CSP4	ATGGTGTCGTTCCTATTAGT	CTAAGCTTTGGTAATGGGCT
CSP5	ATGAAAGTAACTATTGCAAT	TTACAAAACAATTCCTTCTT
CSP6	ATGGAAATGTGTTTTATTTTG	TTATAATTCGGGAAATCCTT
CSP7	ATGAAGCCGATTTTTCTGGTG	TTATAAAACAATTCCTTCT
CSP8	ATGAACAAATCAGTATTGTTTG	TTACAATGTGTTTTTGTATTC
CSP9	ATGGGGTTTGCAAGATTAA	TTAAGGACTGTTTAAGAATG
CSP10	ATGAAGACGTTTGTCGTTTG	TTATGTACTGATTTCTTTTTCA
CSP11	ATGCAAACATTTCAGCCT	TTAACCGGAATACTGTTT
**RT-qPCR primers**		
CSP1-q	CCTGATGATATATACGTGA	TTGTACTATTTGTGGAAGC
CSP2-q	TTGTGGTTTGTCTGTCTT	ACGTAGCTCTTGATGATT
CSP3-q	TGGCGCTCCGAAAACCTT	ATCGCAATCACCACAACCT
CSP4-q	ATGGTGTCGTTCCTATTAGTTTT	TTTCTTCGAGGTTGATGTTGTC
CSP5-q	GGTTACATCGACTGCCTT	TTTTTGCTCCTTTTTTCT
CSP6-q	CCTAAAAAAAGTCATCCCT	TATAATTTCTCGTCCAGCT
CSP7-q	GAAAAACAAAGAACAGGA	TATTTAGCAGACAACTCG
CSP8-q	CAGACACCTGTACCAAAG	TCATCAAATAACCAATCA
CSP9-q	TAGTAAGCGATTATTGGA	TATGGATAGCCTCAGGTA
CSP10-q	AAGTCCTTCCTGACGCTC	AGTTCCCCATACCAATCG
CSP11-q	ACTCCGCCAACTAAAATG	ACTGACTGCACGAACCCT
RPL19	AAGGAAGGCATTGTGGAT	GACGCAAATCTCGCATAC

**Table 2 insects-13-00183-t002:** Scape, pedicel, and flagellomeres lengths (mean ± SE) of adult *O. communa* antennae.

Antennal Segment	Length (μm)	
	Male	Female
Scape	283.4 ± 12.96	273.6 ± 10.17
Pedicel	158.8 ± 8.31	179.2 ± 7.51
Flagellomere 1	273 ± 4.93	294.6 ± 2.82
Flagellomere 2	230 ± 5.43	229.4 ± 8.66
Flagellomere 3	184.4 ± 2.79	190.8 ± 2.80
Flagellomere 4	202.6 ± 4.75	215.2 ± 9.16
Flagellomere 5	194.6 ± 7.31	198.4 ± 4.41
Flagellomere 6	186.6 ± 11.57	173.0 ± 5.02
Flagellomere 7	183.4 ± 7.05	183.8 ± 7.59
Flagellomere 8	186.8 ± 3.51	178 ± 8.93
Flagellomere 9	276.2 ± 9.84	265.6 ± 6.49
Total length	2359.8 ± 42.87	2381.6 ± 40.57

**Table 3 insects-13-00183-t003:** Lengths, widths, and density of various adult *O. communa* antennal sensilla. + to ++++ indicate relative density of different sensilla.

Sensillum Type	Length (μm)		Width (μm)		Density
	Male	Female	Male	Female	
Sensilla trichodea 1	43.45 ± 0.88	42.63 ± 1.09	2.35 ± 0.09	2.50 ± 0.08	++++
Sensilla trichodea 2	22.28 ± 0.56	22.65 ± 0.48	1.88 ± 0.04	2.11 ± 0.04	+++
Sensilla chaetica	47.07 ± 1.48	48.47 ± 1.85	2.74 ± 0.08	3.02 ± 0.07	+++
Sensilla basiconica 1	11.83 ± 0.21	13.27 ± 1.40	1.45 ± 0.03	1.60 ± 0.03	++
Sensilla basiconica 2	10.85 ± 0.28	10.70 ± 0.27	1.81 ± 0.06	1.80 ± 0.06	++
Sensilla coeloconica	5.03 ± 0.32	4.99 ± 0.32	1.28 ± 0.03	1.24 ± 0.03	+
Sensilla styloconica	3.62 ± 0.19	4.08 ± 0.23	1.13 ± 0.04	1.12 ± 0.05	+
Böhm bristle	7.16 ± 0.43	6.46 ± 0.34	1.30 ± 0.05	1.10 ± 0.09	++

**Table 4 insects-13-00183-t004:** The similarity of 11 *O. communa* CSP amino acid sequences.

	CSP1	CSP2	CSP3	CSP4	CSP5	CSP6	CSP7	CSP8	CSP9	CSP10	CSP11
CSP1		10.69%	10.73%	15.33%	15.33%	15.71%	15.33%	11.11%	11.11%	16.86%	6.04%
CSP2	10.69%		48.18%	21.43%	21.90%	18.98%	23.36%	10.95%	17.52%	21.17%	5.88%
CSP3	10.73%	48.18%		18.12%	26.67%	16.30%	23.70%	11.85%	22.22%	29.63%	5.38%
CSP4	15.33%	21.43%	18.12%		40.44%	38.81%	43.28%	31.85%	30.15%	43.28%	11.43%
CSP5	15.33%	21.90%	26.67%	40.44%		32.58%	41.22%	30.08%	25.56%	44.27%	17.52%
CSP6	15.71%	18.98%	16.30%	38.81%	32.58%		39.53%	21.71%	30.53%	42.64%	17.39%
CSP7	15.33%	23.36%	23.70%	43.28%	41.22%	39.53%		19.38%	31.54%	42.19%	7.69%
CSP8	11.11%	10.95%	11.85%	31.85%	30.08%	21.71%	19.38%		15.50%	29.69%	12.32%
CSP9	11.11%	17.52%	22.22%	30.15%	25.56%	30.53%	31.54%	15.50%		37.90%	11.11%
CSP10	16.86%	21.17%	29.63%	43.28%	44.27%	42.64%	42.19%	29.69%	37.90%		7.01%
CSP11	6.04%	5.88%	5.38%	11.43%	17.52%	17.39%	7.69%	12.32%	11.11%	7.01%	

## Data Availability

The datasets generated for this study can be found in the GenBank using accession number: OL456274-OL456284.

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
