# Peer review of "Scanning Electron Microscopic Analysis of Antennal Sensilla and Tissue-Expression Profiles of Chemosensory Protein Genes in Ophraella communa (Coleoptera: Chrysomelidae)"

_insects, 2022, doi:10.3390/insects13020183_

Round 1

Reviewer 1 Report

     The reviewer has read expectantly the second revised manuscript entitled as “Scanning Electron Microscopic Analysis of Antennal Sensilla and Tissue-Expression Profiles of Chemosensory Proteins Genes in Ophraella communa (Coleoptera: Chrysomelidae)” written by Dr. Chao Ma et al. The authors revised considerably the manuscript along the reviewer’s suggestions and the manuscript has been significantly improved. However, as some problems are remaining, the reviewer will add some comments in order to improve the manuscript more.

Line 17              morphology of sensilla on the antennae was examined by scanning electron microscopy -> morphology of sensilla on the antennae were examined by scanning electron microscopy

Line35               CSPs may be related to function in O. communa. -> (Sorry, the reviewer’s mistake) CSPs may be related to olfactory function in O. communa.

Lines 45-46       geographical -> Concretely speaking, what kind of stimuli do the sensilla respond to?

Line 53                  (CSPs) have extremely high concentrations in sensilla lymph, ->         (CSP) are extremely concentrated in sensilla lymph,

Line 55              CSPs are small, highly soluble proteins -> CSPs are small and highly soluble proteins

Line 57              and pheromones to odorant receptor -> and pheromones to given olfactory receptors

Line 65              is an herbivorous insect -> is a herbivorous insect

Line 95              According to the pictures taken, -> using the pictures taken,

Lines 95-96       we also estimated the overall distribution density of each sensilla. -> (Reviewer’s opinion) How did the authors estimate the distribution density? Please write the estimation method.

Line 99              while -> Please omit the word ‘while’.

Line 154            United States -> USA

Line 155            United States -> USA

Line 168            ST were the most frequent sensilla type and were attached to all-> ST were the most frequent type of sensilla and were found in all

Table 3              (Reviewer’s opinion) Though density of all types of sensilla is shown in this table, the density marks (+) are not defined. Please show the definition for each mark.

Figure 2D       (Reviewer’s opinion) Though the tips of ST1 are shown in Figure 2D, it is difficult to evaluate whether or not the terminal pore is present by this figure. Please add a more magnified micrograph if possible.

                        (Reviewer’s opinion) The socket of a S. trichodium 1 appears flexible but that of a S. trichodium 2 does not so. These morphological feature coincides with the structural difference of side wall of the shafts of these two types of Sensilla.

Line 180         (B) Scattered pores distributed around -> (B) Pores were scattered around

Line 182         (H) Magnification of curved wall of SCH. -> (H) Magnification of side wall of SCH.

Line 189         pores and declined at 30° angles relative to the cuticle surface (Figure 2E). -> pores and ST2 declined at 30° angles relative to the cuticle surface (Figure 2E).

Line 195         curved walls with no pores and were set in wide sockets at 60° angles relative to the cuticle -> curved shafts with no pores on the side wall and were set in wide sockets at 60° angles relative to the cuticle

Line 207         and a few SB1 had a hole near their bottoms (Figure 3B, 3C). > (Reviewer’s opinion) A hole is not visible in Figure 3C.

Figure 3             (Reviewer’s opinion) Figure 3d, f, g are good but SST in Figure 3I is probably a broken one. How many times did the authors find these sensilla which have a terminal pore similar to the pore in Figure 3I?

                          (Reviewer’s opinion) Images of Sensilla SB1, SB2 and SCO in Figure 3E are too small to reveal characteristics of these sensilla.

Line 208            pores were distributed on wall of SB1 (Figure 3D). -> pores were distributed on shaft side wall of SB1 (Figure 3D).

Lines 221-222   The top parts of flagellomeres displayed centrally distributed pit areas of SB2. -> (Reviewer’s opinion) The reviewer cannot understand this sentence.

Lines 226-227   The appearance of the SCO can be likened to pointed nails sitting on raised sockets on the antennal surface (Figure 3E). -> (Reviewer’s opinion) The features written here cannot be visible in Figure 3E. More magnified micrographs are needed.

Line 231          which displayed centrally distributed areas of SB2 (Figure 3E). -> (Reviewer’s opinion) The reviewer cannot understand the phrase.

Lines 237-238 They had a multi-pores wall and there was a hole on their tips (Figure 3H, 237 3I). -> (Reviewer’s opinion) Tiny pores (possibly olfactory pores) of the sensillar side wall are not visible in Figure 3I. If the contrast of the micrograph will be strengthen, the tiny pores may become visible.
How many sensilla having such apical hole did the authors observe in this work? The reviewer disputes whether the apical hole shown in this Figure3I is normal and guesses that it is a damaged apex of the sensillar shaft.

Line 244          declined by 60°–90°, relative to the cuticle surface -> declined by 60°–90° relative to the cuticle surface

Line 254          Based on previous -> Based on the previous

Figure 5 is more visible than the previous one. It is easy to read characters on the both axes.

Line 285          they were particularly dense on the nineth flagellomere -> (Reviewer’s opinion) The sensillar densities on each flagellomere are not described in RESULTS at all. If the authors want to discuss these, they should show the data in RESULTS. Or please replace these description with a weaker expression. For example, they appeared dense on the ninth flagellomere.

Line 299          Based on previous studies, the pore may secrete pheromone to promote mating -> (Reviewer’s opinion) The reviewer does not think that these pores secrete pheromone. Do the authors really think so?

Lines 299-300 have enzymatic activity degrading molecules -> (Reviewer’s opinion) The adequate subject is not found in this sentence.

Lines 304-305     SCH are the longest sensilla type on the O. communa antennal surface -> SCH are the longest sensilla on the O. communa antennal surface

Line 311          SCH are generally regarded as tactile sensation receptors. -> SCH are generally regarded as tactile receptors.

Line 313          their density increased at the distal portion of flagellomere 4–9 -> (Reviewer’s opinion) The sensillar densities on each flagellomere are not described in RESULTS at all. If the authors want to discuss these, they should show the data in RESULTS.

Lines316-317  findings were similar to the findings for O. nipae and T. fuscum and less than the findings for C. aeruginosa, T. castaneum, and P. striolata -> (Reviewer’s opinion) The subject ‘findings’ is ambiguous and inadequate.

Line 320          The types number of SB found in O. communa was > The number of SB found in O. communa was

Line 321          this may be because O. communa is an oligophagous insect -> this may arise from the fact that O. communa is an oligophagous insect

Line 325          a centrally distributed SB2 pit area on the top part of flagellomeres -> (Reviewer’s opinion) The reviewer cannot understand this phrase.

Line 326          P. striolata have previously been found to have sunken regions -> (Reviewer’s opinion) The word ‘previously’ cannot be used in the sentence of the present perfect form.

Lines339-340      Böhm bristles were found to be distributed on the articulation between the head and scape (Figure 4A), and the scape and pedicel (Figure 4B), > Böhm bristles were found to be distributed on the articulations between the head and scape (Figure 4A) and between the scape and pedicel (Figure 4B),

Author Response

Response to Reviewer Comments

Response to Reviewer 1:

The reviewer has read expectantly the second revised manuscript entitled as “Scanning Electron Microscopic Analysis of Antennal Sensilla and Tissue-Expression Profiles of Chemosensory Proteins Genes in Ophraella communa (Coleoptera: Chrysomelidae)” written by Dr. Chao Ma et al. The authors revised considerably the manuscript along the reviewer’s suggestions and the manuscript has been significantly improved. However, as some problems are remaining, the reviewer will add some comments in order to improve the manuscript more.

Line 17              morphology of sensilla on the antennae was examined by scanning electron microscopy -> morphology of sensilla on the antennae were examined by scanning electron microscopy

Line35               CSPs may be related to function in O. communa. -> (Sorry, the reviewer’s mistake) CSPs may be related to olfactory function in O. communa.

Lines 45-46       geographical -> Concretely speaking, what kind of stimuli do the sensilla respond to?

Response: We have deleted “geographical”.

Line 53             (CSPs) have extremely high concentrations in sensilla lymph, ->         (CSP) are extremely concentrated in sensilla lymph,

Line 55              CSPs are small, highly soluble proteins -> CSPs are small and highly soluble proteins

Line 57              and pheromones to odorant receptor -> and pheromones to given olfactory receptors

Line 65              is an herbivorous insect -> is a herbivorous insect

Line 95              According to the pictures taken, -> using the pictures taken,

Lines 95-96       we also estimated the overall distribution density of each sensilla. -> (Reviewer’s opinion) How did the authors estimate the distribution density? Please write the estimation method.

Response: Thanks for your comments. Using the pictures taken, we roughly counted the number of various sensilla, and use different numbers of plus signs to represent four densities (+, ++, +++ and ++++). We have revised the sequence in the part of method.

Line 99              while -> Please omit the word ‘while’.

Line 154            United States -> USA

Line 155            United States -> USA

Line 168            ST were the most frequent sensilla type and were attached to all-> ST were the most frequent type of sensilla and were found in all

Table 3              (Reviewer’s opinion) Though density of all types of sensilla is shown in this table, the density marks (+) are not defined. Please show the definition for each mark.

Response: Thanks for your comments. + represent slight, ++ represent less dense, +++ represent dense and ++++ represent extremely dense, respectively. We have revised the sequence in the part of method.

Figure 2D       (Reviewer’s opinion) Though the tips of ST1 are shown in Figure 2D, it is difficult to evaluate whether or not the terminal pore is present by this figure. Please add a more magnified micrograph if possible.

Response: Thanks for your comments. We found that ST1 lay almost parallel to the cuticles, so we cannot shoot tips of ST1.

(Reviewer’s opinion) The socket of a S. trichodium 1 appears flexible but that of a S. trichodium 2 does not so. These morphological feature coincides with the structural difference of side wall of the shafts of these two types of Sensilla.

Response: Thanks for your comments. The results are consistent with what you said. The walls of ST1 had longitudinal grooves, while the side wall of ST2 had densely small pores.

Line 180     (B) Scattered pores distributed around -> (B) Pores were scattered around

Line 182     (H) Magnification of curved wall of SCH. -> (H) Magnification of side wall of SCH.

Line 189         pores and declined at 30° angles relative to the cuticle surface (Figure 2E). -> pores and ST2 declined at 30° angles relative to the cuticle surface (Figure 2E).

Line 195         curved walls with no pores and were set in wide sockets at 60° angles relative to the cuticle -> curved shafts with no pores on the side wall and were set in wide sockets at 60° angles relative to the cuticle

Line 207         and a few SB1 had a hole near their bottoms (Figure 3B, 3C). > (Reviewer’s opinion) A hole is not visible in Figure 3C.

Response: Thanks for your comments. In the present study, Figure 3B mainly showed that a few SB1 had a hole near their bottoms, and Figure 3C mainly showed that SB1 were set in raised circular sockets and declined at 30° angles relative to the cuticle surface. We have revised the sequence in the manuscript.

Figure 3             (Reviewer’s opinion) Figure 3d, f, g are good but SST in Figure 3I is probably a broken one. How many times did the authors find these sensilla which have a terminal pore similar to the pore in Figure 3I?

Response: Thanks for your comments. When the magnification is high, we found all SST had a terminal pore in their tips. Actually, you can see a pore in SST tip in Figure 3H. In addition, what Figure 3I showed is the normal structural of SST.

 (Reviewer’s opinion) Images of Sensilla SB1, SB2 and SCO in Figure 3E are too small to reveal characteristics of these sensilla.

Response: Thanks for your comments. Figure 3E mainly showed that the pit areas centrally distributed of SB2. If the magnification increases, it cannot to see the full pit area in the figure.

Line 208            pores were distributed on wall of SB1 (Figure 3D). -> pores were distributed on shaft side wall of SB1 (Figure 3D).

Lines 221-222   The top parts of flagellomeres displayed centrally distributed pit areas of SB2. -> (Reviewer’s opinion) The reviewer cannot understand this sentence.

Response: Thanks for your comments. The sequence mainly showed the distribution area of SB2. Different from that of SB1 mainly distributed on the surface of antennae, the SB2 is just distributed on the top parts of flagellomeres 4–9. We have revised the sequence in the manuscript.

Lines 226-227   The appearance of the SCO can be likened to pointed nails sitting on raised sockets on the antennal surface (Figure 3E). -> (Reviewer’s opinion) The features written here cannot be visible in Figure 3E. More magnified micrographs are needed.

Response: Thanks for your comments. Figure 3E mainly described the distribution area of SCO (as same as SB2), and Figure 3G, 3H were describe the detail morphology of SCO. We have revised the sequence in the manuscript.

Line 231          which displayed centrally distributed areas of SB2 (Figure 3E). -> (Reviewer’s opinion) The reviewer cannot understand the phrase.

Response: Thanks for your comments. The sequence mainly showed the distribution position of SCO is completely consistent with that of SB2 (Figure 3E). We have revised the sequence in the manuscript.

Lines 237-238 They had a multi-pores wall and there was a hole on their tips (Figure 3H, 237 3I). -> (Reviewer’s opinion) Tiny pores (possibly olfactory pores) of the sensillar side wall are not visible in Figure 3I. If the contrast of the micrograph will be strengthen, the tiny pores may become visible.
How many sensilla having such apical hole did the authors observe in this work? The reviewer disputes whether the apical hole shown in this Figure3I is normal and guesses that it is a damaged apex of the sensillar shaft.

Response: Thanks for your comments. Because SST located in a densely distributed area of SB2, and the number of SST is very small, so SST is often blocked by other sensilla. We have tried our best to shoot SST, as showed in Figure 3I. However, there is no doubt that all SST had a pore in their tips, because we observed a small hole at the top of all SST during our shoot process. Actually, you can see a pore in SST tip in Figure 3H. In addition, what Figure 3I showed is the normal structural of SST.

Line 244          declined by 60°–90°, relative to the cuticle surface -> declined by 60°–90° relative to the cuticle surface

Line 254          Based on previous -> Based on the previous

Figure 5 is more visible than the previous one. It is easy to read characters on the both axes.

Line 285          they were particularly dense on the nineth flagellomere -> (Reviewer’s opinion) The sensillar densities on each flagellomere are not described in RESULTS at all. If the authors want to discuss these, they should show the data in RESULTS. Or please replace these description with a weaker expression. For example, they appeared dense on the ninth flagellomere.

Response: Thanks for your comments. We have revised the sequence in the manuscript.

Line 299          Based on previous studies, the pore may secrete pheromone to promote mating -> (Reviewer’s opinion) The reviewer does not think that these pores secrete pheromone. Do the authors really think so?

Response: Thanks for your comments. We have revised the sequence in the manuscript.

Lines 299-300 have enzymatic activity degrading molecules -> (Reviewer’s opinion) The adequate subject is not found in this sentence.

Response: Thanks for your comments. We have revised the sequence in the manuscript.

Lines 304-305     SCH are the longest sensilla type on the O. communa antennal surface -> SCH are the longest sensilla on the O. communa antennal surface

Line 311          SCH are generally regarded as tactile sensation receptors. -> SCH are generally regarded as tactile receptors.

Line 313          their density increased at the distal portion of flagellomere 4–9 -> (Reviewer’s opinion) The sensillar densities on each flagellomere are not described in RESULTS at all. If the authors want to discuss these, they should show the data in RESULTS.

Response: Thanks for your comments. we have revised the sequence and add the description in RESULTS part.

Lines316-317  findings were similar to the findings for O. nipae and T. fuscum and less than the findings for C. aeruginosa, T. castaneum, and P. striolata -> (Reviewer’s opinion) The subject ‘findings’ is ambiguous and inadequate.

Response: Thanks for your comments. We have revised the sequence in the manuscript.

Line 320          The types number of SB found in O. communa was > The number of SB found in O. communa was

Line 321          this may be because O. communa is an oligophagous insect -> this may arise from the fact that O. communa is an oligophagous insect

Line 325          a centrally distributed SB2 pit area on the top part of flagellomeres -> (Reviewer’s opinion) The reviewer cannot understand this phrase.

Response: Thanks for your comments. We have revised the sequence in the manuscript.

Line 326          P. striolata have previously been found to have sunken regions -> (Reviewer’s opinion) The word ‘previously’ cannot be used in the sentence of the present perfect form.

Response: Thanks for your comments. We have deleted “previously”

Lines339-340      Böhm bristles were found to be distributed on the articulation between the head and scape (Figure 4A), and the scape and pedicel (Figure 4B), > Böhm bristles were found to be distributed on the articulations between the head and scape (Figure 4A) and between the scape and pedicel (Figure 4B),

Response: we are grateful to receive your detailed suggestions. We have carefully revised the manuscript according to your advice, and the revised parts have been marked in red. This greatly improves the level of our article. Thanks again.

Reviewer 2 Report

I consider that this version was improved and is ready for publication.

Author Response

Thanks for your comments.

This manuscript is a resubmission of an earlier submission. The following is a list of the peer review reports and author responses from that submission.

Round 1

Reviewer 1 Report

The reviewer has read with interest the manuscript entitled as “Scanning Electron Microscopic Analysis of Antennal Sensilla and Tissue-Expression Profiles of CSP Genes in Ophraella communa” written by Dr. Chao Ma et al. and submitted to “Insects”. The authors studied the antennal sensilla with SEM and CSP genes with molecular biological techniques, and obtained some interesting results. However, the manuscript includes some serious problems as follows.

  1. The relation between the morphological study of antennal sensilla and the molecular biological study of CSP genes appears to be not clear. These studies should be more associated with each other.
  2. Most of SEM micrographs in this manuscript are too low magnified to identify types of sensilla and to infer the functions of sensilla. In order to identify the sensillar types and to infer the sensillar function from SEM micrographs, it is necessary to find some specific characteristics in sensillar external structures. The reviewer would ask the authors to obtain higher magnified micrographs enough to observe these structures.
  3. Though the number of each type of sensilla was not counted in this study, the numbers were dealt in the argument on the sensillar function in Discussion. The authors should count the sensilla, if the sensillar numbers are dealt in relation to sensillar function and species comparison in Discussion.
  4. Although the authors showed the similarity of 11CSP amino acid sequences, the authors did not discuss on the sequences at all. What do the authors indicate by the similarity comparison?
  5. The study on CSPs is not enough connected to the morphological study on antennal sensilla. If an immunostaining method will be used in addition, some CSPs may be linked to particular types of sensilla.
  6. The reviewer does not comment on a body text at this time and will comment on the revised version of the text which the authors will make.

Author Response

Thank you for your useful guidance and advice. I have revised the manuscript according to your suggestions. Thank you very much.

Reviewer 2 Report

The manuscript describes the antennal sensilla found in males and females of a phytophagous beetle as well as the expression of 11 predict chemosensory proteins in some organs, including antennae. The research is interesting with good experimental design and results of good quality are well discussed. I have some minor concerns labeled in the attached file to improve the manuscript that deserves publication.

Author Response

(The authors gave the same response as above.)

Round 2

Reviewer 1 Report

General Comments to Authors

The reviewer has read with sinking heart the revised manuscript entitled as “Scanning Electron Microscopic Analysis of Antennal Sensilla and Tissue-Expression Profiles of Chemosensory Proteins Genes in Ophraella communa (Coleoptera: Chrysomelidae)” written by Dr. Chao Ma et al. The authors mildly revised the manuscript along the reviewer’s suggestions. Thus the reviewer will state additional comments.

  1. The morphological study of antennal sensilla seems to be scarcely linked to the molecular biological study of CSP genes. In order to link between them, for example, it is useful to determine whether or not some CSPs such as CSP2, CSP4, and CSP6 are binding proteins for some volatile components of egg-laying plants.
  2. Most of SEM micrographs in this manuscript are too low magnified to identify types of sensilla and to infer the functions of sensilla. Generally higher magnified and high-quality micrographs are necessary to classify the sensilla and identify the function of sensilla. As the authors are especially interested with olfaction, the olfaction-specific structures should be observed. These structures are observable only in high-magnified and high-resolution micrographs. As the SEM (Hitachi S-3400N) used by the authors has resolving power of 10 nm at 3kV according to a specification sheet of S-3400N, it is capable for the authors to get more high quality micrographs.
    In addition, s. trichodea and s. chaetica were often confused in low magnified SEM images in some previous papers. However, as many olfactory pores are present on side wall in s. trichodea and longitudinal stout grooves are present on side wall in s. chaetica, it is not difficult to distinguish correctly between them by getting high magnified images.
  3. The authors stated in “Response to reviewer comments” that authors just compare the number of sensillar types, but not compare the number of sensilla. However they compared the sensillar numbers in some sensillar types in Discussion. It is not scientific to discuss in impressionistic manner.

Minor Comments

Line 26              these insects play > this insect plays

Line 29              using > by

Line 36              may have an olfactory function > may be related to function

Line 46             Geo- > What does “geo” specifically mean?

Lines 56-58       This part of sentences seems to mean that the CSPs are synonymous with binding proteins. Please change the sentences.

Lines 96-97       at least ten sensilla > at least ten sensilla for all types of sensilla

Line 113            thoraces > The authors use “thoraces” here but use “thoraxes” in Figure 5. Please unify these.

Line 135           DNAMAN > show maker and country.

Line 138            a scape, pedicel, and flagellum > a scape, a pedicel, and a flagellum

Figure 1             The SEM images in B, C, and D are too small. Please replace them

Table 2              Table 2 is better to be moved to the next page.

Line 147            Sensilla trichoidea > Sensilla trichodea. It is better to use the same word. The same were found in Lines152 and 165.

Line 149            and bent shafts. > and bent shafts (Figure 2A).

Line 155            horizontal grooves > longitudinal grooves

Line 155            some scattered cuticular pores were distributed > some cuticular pores were scattered

Figure 2             As there are no high-magnified images in this figure, the external structures of sensilla trichodea 1 and 2 are not observable at all. Please replace the SEM images in the figure with high-magnified and clear ones.

Line 160            Details of sensilla trichodea > SEM micrographs of sensilla trichodea

Line 164            Table 3. Lengths, widths and density of various adult O. communa antennal sensilla. > Table 3. Lengths and widths of various adult O. communa antennal sensilla.

                          In this figure there is no data about sensillar density.

Line 168           subtended > declined

Lines 172-173   SCH had a blunt tip and were slightly longer and straighter than ST1. SCH had deep curved walls with no pores > As the SEM micrographs are low magnified ones in Figure 2, it is difficult to observe fine structure of SCH (sensillar tip and pores on side wall) in Figure 2. Please replace them with high-magnified and clear ones.

Line 174            scattered pores were distributed next to > pores were scattered next to

Line 183            SB1 were the longer and thinner SB subtype > SB1 belonged to the longer and thinner SB subtype

Line 184            subtended > declined

Line 186            scattered pores were distributed next to the SB1 > pores were scattered next to the SB1

Figure 3             In Figure 3A, B and D, fine external structures of most sensilla are invisible. Please replace these figures with higher magnified and clear ones.

Line 195            smooth walls > It is difficult to evaluate the fine external structure by Figure 3.

Line 196            subtended > declined

Lines 197-198   a bend was displayed in SB2, which was either considerable or slight > the reviewer cannot understand the sentence.

Line 203            The appearance of the SCO can be likened to pointed nails sitting on raised sockets > Do the authors know the definition of s. coeloconica? It is “the sensilla resembling to cone rising from pit”.

Line 206            flower bone > What is the flower bone?  The reviewer failed to find the word in Dictionaries.

Line 207            on the top of flagellomeres 4–9 > on the most distal part of flagellomeres 4–9,

Line 208           centrally distributed areas > where are the centrally distributed areas?

Line 213            They had smooth walls with pores on their blunt tips (Figure 3D). > As fine external structures of SB are invisible, the readers probably do not approve the description.

Line 214            The SST number was very small, and their distribution was found to be identical to that of SCO. > How many SST and SCO were found on a single flagellomere? If these sensilla were not counted, the sentence is inadequate.

Lines 217-222   The reviewer also agrees that Böhm bristles have morphological features which the authors stated. However, it is better to show high-magnified SEM micrographs, because the readers want to know the facts per se.

Line 219            pointed outward by> declined by

Line 220           (Figure 4A, 4 B) > (Figures 4A, 4 B)

Figure 5             Graphs in Figure 5 are too small to read the characters on horizontal and vertical axes. Please replace Figure 5 with a lager one.

Line 245            expression fold changes > Please show the meaning of this phrase.

Line 235            insect CSPs are involved > some insect CSPs are involved

Line 255            distribution of six antennae sensilla types > distribution of six types of antennae sensilla

Lines 258-261   In this study, as the number and density of distributed sensilla were not measured, the comparison of the number and density with those of other species is inadequate. Similar problems are found in Lines 288-293.

Lines 267-270   In the reviewer’s point of view, the pores scattered around the ST sockets are not related with olfaction and are probably orifices of secretion glands (Weis A et al. 1999 Exocrine glands in the antennae of the carabid beetle, Platynus assimilis (Paykull) 1790 (Coleoptera, Carabidae, Pterostichinae. International J Insect Morph Embyol 28 331-335)

Lines 296-297   Generally, these sensilla are regarded as sensors that recognize humidity and temperature changes. > This is not true. The hygro- and thermoreceptive sensilla of cockroach, locust, stick insect, honey bee etc. were examined extensively with electrophysiological methods and those of Drosophila were examined with molecular biological methods. These studies did not show the hygro- and thermoreceptive sensilla generally belong to SCO or SST.  

Line 304            CSPs are an ancient and highly conserved protein family. > CSPs belong to an ancient and highly conserved protein family.

Line 306 that CSPs have high concentrations > that some CSPs have high concentrati

Author Response

Response to Reviewer Comments

Response to Reviewer 1:

The reviewer has read with sinking heart the revised manuscript entitled as “Scanning Electron Microscopic Analysis of Antennal Sensilla and Tissue-Expression Profiles of Chemosensory Proteins Genes in Ophraella communa (Coleoptera: Chrysomelidae)” written by Dr. Chao Ma et al. The authors mildly revised the manuscript along the reviewer’s suggestions. Thus the reviewer will state additional comments.

  1. The morphological study of antennal sensilla seems to be scarcely linked to the molecular biological study of CSP genes. In order to link between them, for example, it is useful to determine whether or not some CSPs such as CSP2, CSP4, and CSP6 are binding proteins for some volatile components of egg-laying plants.

Response: Thanks for your comments. Later we will study the function and immunostaining about CSP in sequential researches. If the reviewer still consider the biological study of CSPs in this study is inappropriate, we can delete the research part on CSPs.

  1. Most of SEM micrographs in this manuscript are too low magnified to identify types of sensilla and to infer the functions of sensilla. Generally higher magnified and high-quality micrographs are necessary to classify the sensilla and identify the function of sensilla. As the authors are especially interested with olfaction, the olfaction-specific structures should be observed. These structures are observable only in high-magnified and high-resolution micrographs. As the SEM (Hitachi S-3400N) used by the authors has resolving power of 10 nm at 3kV according to a specification sheet of S-3400N, it is capable for the authors to get more high quality micrographs.
    In addition, s. trichodea and s. chaetica were often confused in low magnified SEM images in some previous papers. However, as many olfactory pores are present on side wall in s. trichodea and longitudinal stout grooves are present on side wall in s. chaetica, it is not difficult to distinguish correctly between them by getting high magnified images.

Response: Thanks for your comments. Based on the magnification function of SEM (Hitachi S-3400N), although we did not enlarge sensilla at 3 Kv, we can identify the unique structure and organization of different sensilla. For example, the reviewer thought there were many olfactory pores on side wall in s. trichodea, while we just observed grooves. In addition, some studies showed that there were no pores on s. trichodea in other chrysomelidae insects. This is because different insects have different sensilla structures. Reference were list below: Zhang, G. H. , Li, B. L. , & Li, C. R. . (2016). Morphology and distribution of antennal sensilla of female phyllotreta striolata (fabricius) (coleoptera: chrysomelidae). Zhang, L. , Ren, L. L. , Luo, Y. Q. , & Zong, S. X. . (2013). Scanning electron microscopy analysis of the cephalic sensilla of chrysolina aeruginosa fald. (coleoptera, chrysomelidae); Bartlet, E. , Romani, R. , Williams, I. H. , & Isidoro, N. . (1999). Functional anatomy of sensory structures on the antennae of psylliodes chrysocephala l. (coleoptera: chrysomelidae).

  1. The authors stated in “Response to reviewer comments” that authors just compare the number of sensillar types, but not compare the number of sensilla. However they compared the sensillar numbers in some sensillar types in Discussion. It is not scientific to discuss in impressionistic manner.

Response: Thanks for your comments. We reanalyzed the pictures, and we added the distribution density of various sensilla in Table 3.

Minor Comments

Line 26              these insects play > this insect plays

Line 29              using > by

Line 36              may have an olfactory function > may be related to function

Line 46             Geo- > What does “geo” specifically mean?

Response: Geo- means Geographical. We have revised the sequence in line 45.

Lines 56-58       This part of sentences seems to mean that the CSPs are synonymous with binding proteins. Please change the sentences.

Lines 96-97       at least ten sensilla > at least ten sensilla for all types of sensilla

Line 113            thoraces > The authors use “thoraces” here but use “thoraxes” in Figure 5. Please unify these.

Line 135           DNAMAN > show maker and country.

Line 138            a scape, pedicel, and flagellum > a scape, a pedicel, and a flagellum

Figure 1             The SEM images in B, C, and D are too small. Please replace them.

Response: In Figure 1, we mainly emphasize the antennae, including the scape, pedicel, and flagellomeres.

Table 2              Table 2 is better to be moved to the next page.

Response: In Figure 1, we have moved the Table 2 to next page.

Line 147            Sensilla trichoidea > Sensilla trichodea. It is better to use the same word. The same were found in Lines152 and 165.

Line 149            and bent shafts. > and bent shafts (Figure 2A).

Line 155            horizontal grooves > longitudinal grooves

Line 155            some scattered cuticular pores were distributed > some cuticular pores were scattered

Figure 2             As there are no high-magnified images in this figure, the external structures of sensilla trichodea 1 and 2 are not observable at all. Please replace the SEM images in the figure with high-magnified and clear ones.

Response: thanks for your comments. In this study, we mainly focus on the unique features of sensilla surface, such as the grooves on the surface of sensilla trichodea and sensilla chaetica. If the quality of these pictures can't meet your requirements, we need to make an appointment for the instrument to take photos again. It will take some time.

Line 160            Details of sensilla trichodea > SEM micrographs of sensilla trichodea

Line 164            Table 3. Lengths, widths and density of various adult O. communa antennal sensilla. > Table 3. Lengths and widths of various adult O. communa antennal sensilla.

                          In this figure there is no data about sensillar density.

Line 168           subtended > declined

Lines 172-173   SCH had a blunt tip and were slightly longer and straighter than ST1. SCH had deep curved walls with no pores > As the SEM micrographs are low magnified ones in Figure 2, it is difficult to observe fine structure of SCH (sensillar tip and pores on side wall) in Figure 2. Please replace them with high-magnified and clear ones.

Response: thanks for your comments. In this study, we mainly focus on the unique features of sensilla surface, such as the grooves on the surface of sensilla trichodea and sensilla chaetica. If the quality of these pictures can't meet your requirements, we need to make an appointment for the instrument to take photos again. It will take some time.

Line 174            scattered pores were distributed next to > pores were scattered next to

Line 183            SB1 were the longer and thinner SB subtype > SB1 belonged to the longer and thinner SB subtype

Line 184            subtended > declined

Line 186            scattered pores were distributed next to the SB1 > pores were scattered next to the SB1

Figure 3             In Figure 3A, B and D, fine external structures of most sensilla are invisible. Please replace these figures with higher magnified and clear ones.

Response: thanks for your comments. In this study, we mainly focus on the unique features of sensilla surface, such as the grooves on the surface of sensilla trichodea and sensilla chaetica. If the quality of these pictures can't meet your requirements, we need to make an appointment for the instrument to take photos again. It will take some time.

Line 195            smooth walls > It is difficult to evaluate the fine external structure by Figure 3.

Response: thanks for your comments. In this study, we mainly focus on the unique features of sensilla surface, such as the grooves on the surface of sensilla trichodea and sensilla chaetica. If the quality of these pictures can't meet your requirements, we need to make an appointment for the instrument to take photos again. It will take some time.

Line 196            subtended > declined

Lines 197-198   a bend was displayed in SB2, which was either considerable or slight > the reviewer cannot understand the sentence.

Line 203            The appearance of the SCO can be likened to pointed nails sitting on raised sockets > Do the authors know the definition of s. coeloconica? It is “the sensilla resembling to cone rising from pit”.

Response: Thanks for your comments. Based on previous researches, we followed the sensilla terminology provided by Zhang et al. and Zhang et al. to classify sensilla types.

Line 206            flower bone > What is the flower bone?  The reviewer failed to find the word in Dictionaries.

Line 207            on the top of flagellomeres 4–9 > on the most distal part of flagellomeres 4–9,

Line 208           centrally distributed areas > where are the centrally distributed areas?

Line 213            They had smooth walls with pores on their blunt tips (Figure 3D). > As fine external structures of SB are invisible, the readers probably do not approve the description.

Response: thanks for your comments. Actually, in Figure 3D, the pore on SST tips was visible. If the quality of these pictures can't meet your requirements, we need to make an appointment for the instrument to take photos again. It will take some time.

Line 214            The SST number was very small, and their distribution was found to be identical to that of SCO. > How many SST and SCO were found on a single flagellomere? If these sensilla were not counted, the sentence is inadequate.

Response: Thanks for your comments. We reanalyzed the pictures, and we added the distribution density of various sensilla in Table 3.

Lines 217-222   The reviewer also agrees that Böhm bristles have morphological features which the authors stated. However, it is better to show high-magnified SEM micrographs, because the readers want to know the facts per se.

Response: thanks for your comments. In this study, we mainly focus on the unique features of sensilla surface, such as the grooves on the surface of sensilla trichodea and sensilla chaetica. If the quality of these pictures can't meet your requirements, we need to make an appointment for the instrument to take photos again. It will take some time.

Line 219            pointed outward by> declined by

Line 220           (Figure 4A, 4 B) > (Figures 4A, 4 B)

Figure 5             Graphs in Figure 5 are too small to read the characters on horizontal and vertical axes. Please replace Figure 5 with a lager one.

We have remade Figure 5 and enlarged horizontal and vertical axes.

Line 245            expression fold changes > Please show the meaning of this phrase.

Response: When we analyzed the data of RT-qPCR, we set the CSP gene expression level in the thoraxes as the standard, so the detail the specific value represents the expression of the gene in other tissues relative to the thoraxes.

Line 235            insect CSPs are involved > some insect CSPs are involved

Line 255            distribution of six antennae sensilla types > distribution of six types of antennae sensilla

Lines 258-261   In this study, as the number and density of distributed sensilla were not measured, the comparison of the number and density with those of other species is inadequate. Similar problems are found in Lines 288-293.

Response: Thanks for your comments. We have added the relative density of various sensilla in Table 3.

Lines 267-270   In the reviewer’s point of view, the pores scattered around the ST sockets are not related with olfaction and are probably orifices of secretion glands (Weis A et al. 1999 Exocrine glands in the antennae of the carabid beetle, Platynus assimilis (Paykull) 1790 (Coleoptera, Carabidae, Pterostichinae. International J Insect Morph Embyol 28 331-335)

Response: Thanks for your comments. We have revised the part of pores in Discussion and cited related reference. 

Lines 296-297   Generally, these sensilla are regarded as sensors that recognize humidity and temperature changes. > This is not true. The hygro- and thermoreceptive sensilla of cockroach, locust, stick insect, honey bee etc. were examined extensively with electrophysiological methods and those of Drosophila were examined with molecular biological methods. These studies did not show the hygro- and thermoreceptive sensilla generally belong to SCO or SST.  

Response: Thanks for your comments. We have revised the sequence in Discussion and cited related reference.

Line 304    CSPs are an ancient and highly conserved protein family. > CSPs belong to an ancient and highly conserved protein family.

Line 306 that CSPs have high concentrations > that some CSPs have high concentrati

Response: we are grateful to receive your detailed suggestions. We have carefully revised the manuscript according to your advice, and the revised parts have been marked in red. This greatly improves the level of our article. Thanks again.
